# The Effects of Peer Competition-Induced Anxiety on Massive Open Online Course Learning: The Mediating Role of the Behavioral Inhibition System

**DOI:** 10.3390/bs14040324

**Published:** 2024-04-14

**Authors:** Cui Liu, Mengzhen Fang, Min Wang, Yifang Wu, Wen Chen, Yahua Cheng

**Affiliations:** 1College of Teacher Education, Ningbo University, Ningbo 315211, China; 2111041042@nbu.edu.cn (C.L.); 2111041219@nbu.edu.cn (M.F.); 2111041236@nbu.edu.cn (M.W.); 2Collaborative Innovation Center of Assessment toward Basic Education Quality, Beijing Normal University, Beijing 100875, China; 201931630014@mail.bnu.edu.cn; 3Center of Group Behavior and Social Psychological Service, Ningbo University, Ningbo 315211, China; 4State Key Laboratory of Cognitive Neuroscience and Learning, Beijing Normal University, Beijing 100875, China; 5School of Government, Shanghai University of Political Science and Law, Shanghai 201701, China; chengyahua@shupl.edu.cn

**Keywords:** competition, learning performance, micro-lectures, attention

## Abstract

With the increased emphasis on competition in academic settings, anxiety is becoming more common, which inevitably has some impact on students’ learning processes and results. This study aimed to explore how competition-induced anxiety influences students’ subjective cognitive load (SCL), attention levels, and test scores. We also investigated the mediating role of the behavioral inhibition system/behavioral activation system (BIS/BAS) in those factors. A total of 101 college students were recruited in Study 1 to learn from five micro-lectures from massive open online courses (MOOCs) under competitive and non-competitive conditions. The results showed that participants’ state anxiety (SA) scores were higher after the experiment, participants under the competition condition had higher test scores, and the relationship between SA/ trait anxiety (TA) and SCL could be mediated by the BIS. To obtain more objective data on learning processes (attention levels), we conducted Study 2, which collected behavioral and EEG data from 42 college students during the online learning. The results showed that the competition group had higher SA, lower attention levels, and worse test scores, and the relationship between SA/TA and attention levels could be mediated through the BIS. The present study not only expands previous research by finding that BIS functioning plays an important role in the effects of anxiety on cognitive load and attention but also offers implications for using competitive strategies to motivate students according to their aptitudes.

## 1. Introduction

Although individuals in some cultures do not have a competitive nature, it is a common phenomenon for people to compete with each other in some areas, such as China, Japan, and the US [1,2]. Even though some studies have shown that competitive environments could have negative impacts on individuals’ mental health, such as leading to issues like self-harm, anxiety, and despair [3], competition has been widely used as a learning strategy in education contexts to enhance students’ motivation and performance [4]. However, how competition influences the learning process and performance is still controversial.

Many studies have focused on the impact of competition on anxiety and learning. While competition is known to significantly heighten students’ anxiety levels [5,6], how competition impacts the learning process and performance remains controversial [7,8,9]. One perspective suggests that competition impairs students’ learning process [10,11,12]. According to several studies, competition increases individuals’ cognitive loads during the learning process [13,14]. In competitive learning environments, students often opt for easier tasks, leading to decreased levels of learning [15]. Conversely, another viewpoint argues that competition enhances students’ learning outcomes [16,17,18]. Some research has shown a significant positive relationship between competition and academic performance [19]. Researchers have observed that competition motivates students and boosts their performance [20,21]. In addition, some studies have found that competition has no significant impact on students’ learning process [22,23] or their academic achievements [24]. The relationship between competition and learning is intricate, sparking ongoing debates in academia about its effects on students’ learning. Further exploration of the specific mechanisms through which competition influences learning is needed to deepen our understanding of this relationship.

Currently, two primary theories elucidate the impact of competition on the learning process and outcomes. According to Gray, behavior and emotion are regulated by two universal motivational systems: the behavioral activation system (BAS) and the behavioral inhibition system (BIS) [25,26]. The BIS controls unpleasant motivation and anxiety reactions brought on by stimuli associated with anxiety; it can become activated in response to signals of novelty, punishment, and a lack of reward, which causes behavioral inhibition to nudge us away from unpleasant situations. Gray further suggests that the BIS may contribute to the emergence of negative emotional states like anxiety, depression, and sadness. Conversely, the BAS governs appetitive motivation, stimulating behavior and directing us toward desired outcomes [27]. The BAS exhibits high sensitivity to signals of reward, non-punishment, and avoidance of punishment. Gray also posits that the BAS plays a pivotal role in fostering positive emotions such as hope, joy, and pleasure. Although Gray’s theory has previously been widely used to explain maladaptive behaviors such as procrastination and addiction [28], relatively little attention has been paid to the direct effects of the learning process itself, which needs to be explored in terms of how these systems affect cognitive processes, motivation, and overall learning. Moreover, attentional control theory (ACT), viewed through the lens of cognitive processing, elucidates how anxiety influences cognitive performance [29]. ACT has two hypotheses. The first hypothesis holds that anxiety alters the balance between the two attention systems by amplifying the bottom-up stimulus-driven attention system while impeding the top-down goal-directed attention system. The second hypothesis states that anxiety primarily interferes with the inhibitory and transformational functions of the central executive system, reducing its processing efficiency. ACT provides a theoretical basis for researching the effect of anxiety on learning, which is supported by numerous empirical studies [30,31,32]. These studies found that individuals in the high-anxiety group tend to exhibit poorer performance in terms of processing efficiency and task outcomes. Competitive environments often elevate anxiety levels among individuals, leading to decreased attentional control and subsequently influencing learning outcomes.

In essence, both theories have certain strengths. Gray’s theory delves into the mechanisms through which competition impacts learning at a motivational level, while ACT scrutinizes the influence of competition on the learning process from a cognitive perspective. Our study aimed to integrate these two theories to explore how competition impacts learning process and outcomes.

Moreover, on one hand, most of the competition studies in the literature were carried out with a threatening stimulus in laboratory conditions [33,34,35,36], and research on the effect of competition on real learning situations is scarce. On the other hand, a few studies of competition in practical situations have used questionnaires to investigate the impact of daily perceived competition on learning [37,38], and many irrelevant variables are not controlled. Therefore, the present study tried to explore how competition affects MOOC learning, which has seen a surge in popularity globally in recent years [39,40,41,42].

Thus, the current studies aimed to investigate the effect of peer competition on the learning processes and results of students in the setting of MOOC learning. Based on the BIS/BAS hypothesis, Study 1 used experimental research to investigate the effects of peer competition (high and low stress) on trait anxiety (TA) and state anxiety (SA), subjective cognitive load (SCL), and test scores and the role of the BIS/BAS in anxiety and learning. Furthermore, in order to repeat Study 1, we also conducted Study 2, using a portable EEG headband to measure attention levels. These studies specifically examined the learning performance of students from two universities (Beijing Normal University and Ningbo University) in peer competition.

Based on existing research, we hypothesized that competition would increase anxiety and lead to higher subjective cognitive load and lower attention through the BIS. However, the effect of competition on learning performance might be ambiguous, depending on students of different abilities. Specifically, we hypothesized that competition would benefit students with lower ability while harming the performance of students with higher ability.

## 2. Pilot Study

A pilot study was carried out in advance of the formal experiment to evaluate the quiz items’ discrimination and difficulty levels for five micro-lectures. We chose two micro-economics modules and three micro-lectures on physics from “Chinese University MOOC”(https://www.icourse163.org, accessed on 10 September 2019) after searching the resources for suitable content for Chinese students majoring in subjects other than science in their second semester. Each of the micro-lectures was about 5 min long. For the micro-economics students, we chose an easy module which focuses on one concept and a hard module which introduces three concepts and their dynamic relations from one lecturer. Three physics micro-lectures that are comparable in difficulty were chosen from another lecturer, and each one introduces one principle of physics relevant to the Bernoulli principle, Doppler principle, and Pascal’s principle. A quiz was made for every micro-lecture. The knowledge, comprehension, and application questions on the quizzes were based on Bloom’s Taxonomy of Learning Objectives [43]. There were 27 knowledge items, 13 comprehension items, and 6 application items in total among the final 46 items. The three categories were weighted 1:2:3, and the total score ranged from 0 to 71.

We recruited 27 students who did not take part in the formal experiments to watch five micro-lectures and answer fifty related questions. Four poor items were screened out (the item difficulty criterion was *p* < 0.25 or >0.75 for the proportion of students scored correctly; the discrimination criterion was R < 0.20 for the correlation between item and total score). Participants were then asked to answer only 46 items in the formal experiment, which were tested for acceptable reliability (α = 0.610).

## 3. Study 1

### 3.1. Method

#### 3.1.1. Participants

The participants of the experiment were 101 undergraduate and postgraduate students who major in humanities and social sciences (excluding economics) in Ningbo. Fifty of them were assigned to 25 pairs for the competition condition (stressful condition). Each pair of students participated in the experiment together and learned the MOOC materials synchronously. The other 50 students were assigned to the control condition (non-stressful condition), and they completed the experiment by themselves. Students were compensated for this experiment. For the control group, all students were paid RMB 25 after completing the experiment. For the competition group, in addition to the RMB 25, they were told that the one who achieved highest test score would received an extra RMB 5 as a reward. All the students signed informed consent forms after a full explanation of the study procedure. This study was approved by the Institutional Review Board of the Ningbo University.

#### 3.1.2. Materials and Tasks

##### Self-Reported Scales

Behavioral inhibition/activation was assessed using an 18-item self-described measure—“the Behavioral Inhibition/Activation System Scale (BIS-BAS)”—developed by carver and white [27] and revised by Li [44], which has been proved to be suitable for Chinese students. The scale consists of the BIS and BAS, with the BIS consisting of 5 items (e.g., being criticized or blamed makes me feel bad) and the BAS being divided into 3 sub-scales consisting of 5 pleasure seeking items (BASF) (e.g., I often act on impulse), 4 reward response items (BASR) (e.g., winning a race makes me excited), and 4 drive items (BASD) (e.g., I will do everything I can to get what I want). Each participant’s response to the items was rated on a scale of 1 to 4 (1 = “I do not agree at all”, 4 = “I totally agree”). The Cronbach’α coefficient for each dimension of the scale in the study ranged from 0.68 to 0.90.

The State–Trait Anxiety Inventory (STAI) [45] consists of two scales of 40 items. Items 1–20 comprise the State Anxiety Inventory (S-AI), which assesses the patient’s immediate or time-specific anxiety (e.g., I feel calm); items 21–40 comprise the Trait Anxiety Inventory (T-AI), which assesses personality traits (anxiety reactions) and frequent emotional experiences (e.g., I feel happy). Each item is rated on a scale of 1–4, and the total S-AI and T-AI scale scores are calculated separately, with higher scores indicating more severe anxiety. In general, the STAI can be considered reliable and valid, with Cronbach’α coefficients of 0.82 and 0.83 for the 2 sub-scales.

Subjective cognitive load (SCL) was assessed using the PAAS scale [46], which consists of two main items, mental effort and task difficulty, with both measured on a 9-point Likert scale where 1 = very easy or least effort and 9 = very difficult or most effort. SCL was the sum of the two items. Subjects were asked to choose an appropriate number from 1 to 9 according to their feelings after completing the learning task. The Cronbach’α coefficient of the scale was 0.74.

Before the participants were assigned to either the stressful or non-stressful group, they were asked to complete the behavioral inhibition/activation system scale (BIS-BAS scale) and the State–Trait Anxiety Inventory (STAI) items. At the end of experiment, participants were asked to assess their state anxiety with a sub-scale of the STAI (STAI-S) and the SCL of the micro-lecture (V_SCL) and quiz (T_SCL) with the PAAS scale. The V_SCL and T_SCL needed to be completed five times each.

#### 3.1.3. Experimental Procedure

The participants used an online learning system, as shown in Figure 1.

Prior to the experiment, all students were informed that they would watch two micro-lectures on economics and three on physics, and that each video would be followed by a time-limited (2.5 min) quiz. But for the participants who were assigned to the stressful condition, they were told that the person who achieved the highest score would be given an extra reward at the end of the experiment. After the introduction, the students were given a practice trial consisting of a quiz to ensure that they understood the instructions. During the experiment, self-reported cognitive load were collected after each micro-lecture and quiz. The whole experiment took about 40 min.

#### 3.1.4. Data Analysis

IBM SPSS 19.0 was used to analyze the behavioral data. A *t*-test, a repeated measures ANOVA, a one-way ANOVA, and an ANCOVA were conducted to analyze the study outcomes.

### 3.2. Results

#### 3.2.1. Demographic Data

The competition and control group did not differ in terms of gender (*χ*^2^ = 0.001, *p* = 0.971), age (*t* = 1.894, *df* = 99, *p* = 0.061), or years of education (*t* = 1.057, *df* = 99, *p* = 0.293) (Table 1).

#### 3.2.2. Behavioral Results

(1)TA and SA

For the STAI scale, there was no significant group difference for TA (*t* = −1.375, *df* = 98, *p* = 0.172). In terms of SA, the results of the repeated measures ANOVA showed significant differences in measurement time [*F*_(1,98)_ = 61.42, *p* < 0.001, *η*^2^ = 0.385] and, with participants having significantly higher SA post-experiment (41.66 ± 9.75) than before the experiment (34.86 ± 8.07); the interaction between group and measurement time was significant [*F*_(1,98)_ = 5.42, *p* = 0.022, *η*^2^ = 0.052]. Simple effects analysis revealed that the difference in SA between the pre- and post-experiment groups was not significant [*F*_(1,98)_ = 1.363, *p* = 0.246; *F*_(1,198)_ = 1.229, *p* = 0.270], but post-experiment SA was significantly higher than pre-experiment SA in both groups. In particular, this effect was larger in the competition group [competition group: *F*_(1,98)_ = 51.665, *p* < 0.001, *η*^2^ = 0.345; control group: *F*_(1,98)_ = 15.175, *p* < 0.001, *η*^2^ = 0.134] (Figure 2). This suggests that the competitive setting did induce higher levels of SA.

(2)Test Scores

The independent sample *t*-test results showed that the total score of the competition group (43.52 ± 6.57) was significantly higher (*t* = −2.569, *df* = 99, *p* = 0.012) than the control group (39.80 ± 7.89).

(3)Subjective Cognitive Load

Although the competition group and the control group did not reach a statistically significant difference for total SCL in the micro-lectures and quizzes, in general, the control group (12.47 ± 2.45) had a higher SCL than the competition group (12.15 ± 2.48). The repeated measures ANOVA results of SCL showed that the main effect of time was significant (micro-lectures: *F*_(4,95)_ = 9.794, *p* < 0.001, *η*^2^ = 0.292; quizzes: *F*_(4,95)_ = 8.673, *p* < 0.001, *η*^2^ = 0.268), while the main effect of group and the interaction between time and group were not significant (Figure 3A,B). The Bonferroni post hoc test results showed that the SCL values for the second and fifth micro-lecture were significantly higher than those for other micro-lectures (*p* < 0.05), and the SCL values for the second, fourth, and fifth quizzes were higher than those for the first and third quizzes (*p* < 0.05).

(4)Correlations of BIS/BAS with Anxiety, SCL, and Test Scores

Table 2 shows the correlations of the BIS-BAS with anxiety, SCL, and test score. Significant connections existed between the BIS and pre-experiment state anxiety, trait anxiety, and SCL during the quizzes (*r* = 0.319, *p* = 0.001; *r* = 0.365, *p* < 0.001; *r* = 0.260, *p* = 0.009), as well as between BASD and pre-experiment state anxiety and TA (*r* = −0.240, *p* = 0.016; *r* = −0.282, *p* = 0.004).

(5)The Mediating effect of the BIS on the Relationship between Anxiety and SCL

Based on the above correlations, our study found that the BIS may play an important role in the influence of anxiety on SCL. The correlation analyses showed significant correlations between the BIS and TA, SA, and SCL in the quizzes. With the BIS as the mediating variable, SA and TA as the independent variable, and SCL as the dependent variable, the Bootstrap method was used to test the mediating effect of the BIS. If the results of the mediation effect test did not contain 0, the mediating effect was significant. The results of our mediation analysis are shown in Figure 3. The indirect effect values of the BIS in predicting SCL during watching micro-lectures and during participation in quizzes by TA were 0.03, 95%CI = [0.00, 0.06] (Figure 4a); 0.04, 95% CI = [0.01, 0.07] (Figure 4b). The indirect effect of the BIS on the SCL predicted by SA was 0.03, 95% CI = [0.01, 0.06] (Figure 4c).

Study 1 demonstrated the effects of peer competition on the anxiety levels, SCL, and test scores of students at Ningbo University. We found that (a) for SA, the interaction between group and measurement time was significant; (b) compared to the control group, the competition group had better test scores; (c) SCL was not significantly different between the two groups (the competition group and control group); and (d) the BIS plays a mediating role in how TA and SA affect SCL.

## 4. Study 2

Since it is very important for students to focus their attention on the learning task, the level of attention is also one of the important objective indicators of learning effectiveness [47]. We believed conducting Study 2 in a specific organization would enable us to collect objective data (attention level) on students’ learning processes, thereby allowing us to test the relationships among competition, anxiety test score, and attention level in actual MOOC learning. Specifically, Study 2 recorded EEG data on subjects’ attention levels based on Study 1. In addition, Study 1 was conducted used a sample from Ningbo University, but the student population in one area is not representative of the entire student population. It was therefore necessary to determine in Study 2 whether the pattern of results differed in other groups, such as students from another university (Beijing Normal University).

### 4.1. Method

#### 4.1.1. Participants

The participants of the experiment were 42 undergraduate and postgraduate students who major in humanities and social sciences (excluded economics) in Beijing. Twenty-two of them were assigned to 11 pairs for the competition condition (stressful condition). Each pair of students had to participate in the experiment together and learn the MOOC materials synchronously. The other 20 students were assigned to the control condition (non-stressful condition), and they completed the experiment by themselves.

The students were compensated for this experiment. Regarding the control group, all students were paid RMB 60 after completing the experiment. Regarding those in the competition group, they were told that the one who achieved the highest score would receive an extra RMB 20 as a reward. In Study 2, both the control group and the competition group received higher reward amounts than in Study 1, which was due to two main considerations. On the one hand, this is because of the inherent differences in the remuneration of participants between Beijing and Ningbo, as Beijing is the capital city of China and it tends to have higher reward levels than other cities. On the other hand, participants who participate in EEG data collection are also usually owed higher fees. All students signed informed consent forms after a full explanation of the study procedure.

#### 4.1.2. Materials and Tasks

##### Self-Reported Scales

The same self-reported scales used in Study 1 were used in Study 2.

#### 4.1.3. Experimental Procedure

The same materials and procedure used in Study 1 were used in Study 2, but we collected EEG data from students on the basis of Study 1. The students used an online learning system (Figure 1). We used a brainwave-detecting headset to obtain EEG data.

#### 4.1.4. EEG Data Acquisition

The MindSet headset was used to record EEG data in the experiment. This headset includes 7 bands: mid-gamma (41–49.75 Hz), low-gamma (31–39.75 Hz), low-beta (13–16.75 Hz), high-alpha (10–11.75 Hz), low-alpha (7.5–9.25 Hz), theta (3.5–6.75 Hz), and delta (0.5–2.75 Hz). The data were automatically calibrated for ocular artifacts and eye blinks.

#### 4.1.5. Data Analysis

IBM SPSS 19.0 was used to analyze data. Data were analyzed using the same methodology as in Study 1.

The first sixty seconds of the EEG data were eliminated, and the remaining data were averaged at each time point (about one second) before being smoothed using a sliding window that lasted for fifteen seconds (86.67% overlap between successive windows). The average attention level was examined using an ANOVA. The difference in attention level between one time point (a) and the starting period (the first 15 s) of the following time point (b) divided by b is how we finally arrived at the rate of change (ROC) in attention. In other words, ROC = (a − b)/b.

### 4.2. Results

#### 4.2.1. Demographic Data

The competition and control group did not differ in terms of gender (*χ*^2^ = 2.636, *p* = 0.104), age (*t* = 1.891, *df* = 40, *p* = 0.066), or years of education (*t* = 1.472, *df* = 40, *p* = 0.149) (Table 3).

#### 4.2.2. Behavioral Results

(1)TA and SA

For the STAI scale, there was no significant group difference for TA (*t* = −1.335, *df* = 39, *p* = 0.190). In terms of SA, there was no significant group difference in SA before the experiment. While the SA of the competition group (42.19 ± 9.95) was significantly higher (*t* = −2.301, *p* = 0.032) after the experiment than that before the experiment (38.10 ± 8.82), and the SA of the competition group was significantly higher (*t* = −1.619, *p* = 0.046) than that of the control group(38.05 ± 5.77) after the experiment, but for the control group, there was no difference in SA before and after the experiment (Figure 5). The results displayed that the competition condition did induce higher SA.

(2)Test Scores

The total score of the competition group (43.27 ± 5.45) was significantly lower (*t* = 2.117, *df* = 39, *p* = 0.041) than that of the control group (46.70 ± 5.01).

(3)Subjective Cognitive Load

Similar to Study 1, the independent sample *t*-tests of SCL showed that the difference in total SCL, the SCL of watching micro-lectures, and the SCL of partaking in quizzes between the two groups did not reach the level of significance. However, overall, the control group (10.48 ± 2.21) had a lower SCL than the competition group (10.96 ± 2.18). The repeated measures ANOVA results of SCL showed that the main effect of time was significant, while the main effect of group and the interaction between time and group were not significant for both micro-lectures and quizzes (micro-lectures: *F*_(4,34)_ = 6.393, *p* = 0.001, *η*^2^ = 0.429; quizzes: *F*_(4, 34)_ = 3.443, *p* = 0.018, *η*^2^ = 0.288) (Figure 6A,B). The Bonferroni post hoc test results showed that the SCL of the second micro-lecture was significantly higher than that of the other micro-lectures (*p* < 0.05), and the same pattern was found for the SCL in the quizzes (*p* < 0.05).

(4)Attention Level

Attention level was assessed via the EEG index. Our one-way ANOVA showed that the mean attention level of the competition group was significantly lower than that of the control group (*F*_(1,4706)_ = 14.348, *p* < 0.001, *η*^2^ = 0.03). The results of our repeated measures ANOVA showed that the main effect of group was significant for both micro-lectures and quizzes, and the interaction effect of time and group was also significant [micro-lectures: main effect of group: *F*_(4,477)_ = 312.148, *p* < 0.001, *η*^2^ = 0.724; effect of interaction between group and time: *F*_(4,477)_ = 97.431, *p* < 0.001, *η*^2^ = 0.450; test: group main effect: *F*_(4,339)_ = 71.679, *p* < 0.001, *η^2^* = 0.458; effect of interaction between group and time: *F*_(4,339)_ = 50.037, *p* < 0.001, *η^2^* = 0.371] (Figure 7A,B). Pairwise comparisons showed that the attention level of the competition group was significantly lower than that of the control group in the second and third micro-lecture, but that of the competition group was significantly higher than that of the control group the other three times. Furthermore, the competition group had lower attention levels compared to the control group, especially when completing the last quiz. The change pattern regarding attention levels during the quizzes was quite similar for the two groups, except for the fourth quiz, for which the competition group showed lower attention levels while the control group attention showed higher attention levels.

(5)Correlations of BIS/BAS with Anxiety, SCL, Attention Level, and Test Score

Table 4 shows the correlations of the BIS-BAS with anxiety, mean attention level, SCL, and test score. Significant connections existed between the BIS and pre-experiment state anxiety, TA, and mean attention level (*r* = 0.373, *p* = 0.016; *r* = 0.466, *p* = 0.002; *r* = 0.464, *p* = 0.004), as well as between BASD and TA and SCL during the quizzes (*r* = −0.331, *p* = 0.035; *r* = −0.359, *p* = 0.025). The correlation of the BIS with SA and TA was relatively stable in both studies.

(6)The Mediating effect of the BIS on the Relationship between Anxiety and Attention Level

Our study found that the BIS may play an important role in the influence of anxiety on attention level. Correlation analyses showed significant correlations between the BIS and TA, SA, and attention level. With the BIS as the mediating variable, SA and TA as the independent variable, and attention level as the dependent variable, we used the Bootstrap method to test the mediating effect of the BIS. The results of the mediation analysis are shown in Figure 8. The indirect effect values of the BIS in predicting mean attention level by TA was 0.10, 95% CI = [0.03, 0.21]. The indirect effect of the BIS on the mean attention level predicted by SA was 0.09, 95% CI = [0.01, 0.20].

### 4.3. Discussion

Study 2 examined the effects of peer competition on the anxiety levels, SCL, attention levels, and test scores of students from Beijing Normal University. We found that (a) SA was significantly higher in the competition group than in the control group; (b) the competition group showed lower test scores than the control group; (c) SCL was not significantly different between the two groups (the competition group and control group); (d) the average attention level of the competition group was significantly lower than that of the control group; and (e) BIS plays a mediating role in the influence of TA and SA on attention levels.

Although we were interested in assessing the impact of competitive situations on learning processes and performance, participants’ own levels of state and trait anxiety also hindered learning performance, reducing the likelihood of the effects detected by our analysis. Therefore, to provide a more powerful test of the effect of competition on learning, we further analyzed the data for Study 1 and Study 2 in two ways. First, we used the clinical cut-off points for trait and state anxiety. After excluding the results in the sample beyond the clinical cut-off points, the results were consistent with previous ones. Second, we analyzed trait anxiety as a control variable, and the results were still consistent with the results in both studies (more details can be found in the Appendix A), suggesting that the findings were quite stable.

In addition, a MANOVA was also conducted with group as independent variables and SCL and test scores as dependent variables. The results in Study 1 only found a significant group difference in test scores (*F*_(1,99)_ = 6.602, *p* = 0.012, *η*^2^ = 0.063); the total score of the competition group (43.50 ± 1.05) was higher than that of the control group (39.81 ± 1.05). The results in Study 2 found marginally significant group differences in test scores (*F*_(1,37)_ = 6.602, *p* = 0.053, *η*^2^ = 0.097); the total score of the control group (46.69 ± 1.28) was higher than that of the competition group (43.19 ± 1.19). This suggested that the effect of competition on test scores is stable.

## 5. Comparison between Two Universities

It seemed like the effects of competition on the learning process and learning outcomes in Study 1 and Study 2 were inconsistent. In order to further compare the results of the two studies, we also treated university as an independent variable for the analysis. The results are as follows:

(1)TA and SA

For TA, our ANOVA showed that there were no significant university differences, group differences, or differences regarding their interaction. A repeated measures ANOVA revealed the significant main effect of SA measurement time (*F*_(1,137)_ = 40.871, *p* < 0.001, *η*^2^ = 0.230), with post-experiment SA (40.89 ± 0.87) being significantly higher than pre-experiment SA (35.82 ± 0.75); the interaction of measurement time and university was significant (*F*_(1,137)_ = 4.731, *p* = 0.031, *η*^2^ = 0.033), and our simple effects analysis found that the difference in SA between the two universities in pre- and post-experiment was not significant. SA_post was significantly higher than before the experiment in both universities, especially in Ningbo University [Ningbo University: *F*_(1,137)_ = 63.143, *p* < 0.001, *η*^2^ = 0.315; Beijing Normal University: *F*_(1,137)_ = 6.271, *p* = 0.013, *η*^2^ = 0.044]. It was illustrated that the students from Ningbo University generally showed more anxiety compared to the students from Beijing Normal University.

(2)Test Scores

The results of the ANOVA indicated that the main effect of group was not significant (*F*_(1,137)_ = 0.013, *p* = 0.908), the main effect of university was significant (*F*_(1,137)_ = 6.657, *p* = 0.011, *η*^2^ = 0.046), and the test scores of the students from Beijing Normal University (44.86 ± 5.53) were significantly higher than those of the students from Ningbo University (41.94 ± 6.88). The interaction between group and university was also significant (*F*_(1,137)_ = 7.781, *p* = 0.006, *η*^2^ = 0.054). Our simple effects analysis found that Ningbo University’s competitive group performed significantly better than Ningbo University’s control group (*F*_(1,137)_ = 6.171, *p* = 0.014, *η*^2^ = 0.043), while the control group scored higher than the competition group in Beijing Normal University (*F*_(1,137)_ = 2.970, *p* = 0.087, *η*^2^ = 0.021), and the control group from Beijing Normal University scored significantly higher than the control group from Ningbo University (*F*_(1,137)_ = 13.704, *p* < 0.001, *η*^2^ = 0.091). This illustrates the fact that the students at the two universities belong to two groups that are different in nature and that competition affects students’ test scores differently, with competition boosting performance among those from Ningbo University but lowering performance among those from Beijing Normal University.

(3)Subjective Cognitive Load

The results of the ANOVA on the total SCL scores showed a significant main effect for university (*F*_(1,137)_ = 12.563, *p* = 0.001, *η*^2^ = 0.084), with the students from Ningbo University having a significantly higher SCL (12.31 ± 0.24) than those at Beijing Normal University (10.72 ± 0.38). Separate ANOVAs for V_SCL and T_SCL similarly revealed a significant main effect for university only (V_SCL: *F*_(1,137)_ = 13.792, *p* < 0.001, *η*^2^ = 0.091; T_SCL: *F*_(1,137)_ = 10.228, *p* = 0.002, *η*^2^ = 0.069). Again, this illustrates the heterogeneity of the two populations.

## 6. General Discussion

The presented studies explored the effects of peer competition on learning processes and outcomes based on MOOC learning contexts. We found that (a) competitive conditions significantly increased individuals’ SA; (b) there were differences in the effects of competition on test scores, with competition boosting students’ performance in Study 1 but hindering students’ performance in Study 2; (c) although the effect of competition on the SCL of students did not reach a significant level, it showed different trends in the two studies; (d) there were lower levels of attention in the competition group compared to the control group; and (e) the BIS fully mediated between anxiety and SCL in Study 1 and fully mediated between anxiety and attention levels in Study 2.

The current study’s results suggest that competition increases individuals’ state anxiety, and a direct comparison of the data from the two universities also showed that higher SA was induced at Ningbo University compared to Beijing Normal University. Both studies found that competition induces SA in individuals, which is consistent with previous studies [48]. Competitive situations trigger anxiety in individuals [49]. Interestingly, we found that the participants in Study 2 showed lower SA compared to the participants in Study 1. This may be due to the different levels of the universities chosen for the two studies, as Beijing Normal University, as a “double first-class” university, was expected to have smarter students compared to Ningbo University [50]. The effect of competition seems to be dynamic for different individuals. This result is consistent with previous findings that students with high ability are better at coping with competition emotionally and show a lower degree of anxiety than peers with low ability [51,52]. This may be because students with high ability have more confidence to deal with examination tasks, better coping strategies, and a stronger learning ability. Thus, the stress and anxiety they felt during the examination were reduced [53].

The effect of competition on students’ test scores yielded different results in Study 1 and Study 2. Study 1 found that the competitive group had higher test scores than the control group, while Study 2 found that the control group scored higher. A direct comparison revealed that the students in Study 2 scored higher. Why does competition appear to have a different effect on learning outcomes? It may be due to the fact that anxiety may have more severe consequences on the performance of high-achieving students. For example, researchers who examined the relationship between anxiety and test scores in first and second graders in the US found that a negative correlation between anxiety and test scores was only present in students with a high working memory, i.e., the kind of child who has the greatest potential for high achievement [54]. The OECD also reported that, compared to low-ability students, high-ability students’ anxiety is more strongly associated with lower test scores [55]. Another possible explanation is that the competition situation induced different levels of anxiety in the two studies, with the competition group in Study 1 having moderate levels of anxiety. According to the Yerkes–Dodson law, appropriate anxiety or stress can promote and enhance productivity. Given the different effects that competition can have on different groups of students, teachers need to be careful about using this strategy in real classroom situations and use different instructional strategies for different students.

Although there were no significant differences in SCL between the competition and control groups in both Study 1 and Study 2, they exhibited different trends. Direct comparisons also suggested that the students from Ningbo University reported higher cognitive loads. In Study 1, the SCL of the control group was higher than that of the competition group, whereas in Study 2, the SCL reported by the competition group was higher than that of the control group. This paradox can be explained by ACT, which examines learning and problem solving primarily based on attention control. Competition-induced anxiety disrupts the balance between the two attention systems, weakening the goal-driven top-down attention system and enhancing the stimulus-driven bottom-up attention system [29]. In Study 1, the balance of the attention system in the competitive group was broken, and on the contrary, the control group could invest more cognitive effort in the learning process [56], whereas in Study 2, the competitive group had sufficient resources to invest in the learning process due to the relatively low level of anxiety induced by them. A high cognitive load is detrimental to learning [57]; it causes rapid fatigue, reduced flexibility, and frustration and is an important cause of decreased performance [58]. Therefore, we should find ways to reduce students’ load in the learning process so that they can have sufficient cognitive resources to complete learning tasks.

In order to perform optimally in competitive situations, students must learn to cope with competition-induced emotional responses such as anxiety or worry and must focus attention on current task-relevant information [59]. Our EEG data in Study 2 indicated lower levels of student attention under competitive conditions, which is consistent with the consensus of previous studies [59,60,61]. Competition means increased stress and anxiety. According to ACT, anxiety impairs individuals’ attentional control, making them less able to focus on the task at hand and more easily distracted. Future research needs to further investigate how to maintain individual attention levels in competitive situations, which, in turn, can provide more strategies to improve students’ performance in real-world learning situations.

The current study’s findings indicate that in Study 1, SA and TA influenced students’ SCL when watching the micro-lectures and partaking in the quizzes by activating individuals’ BIS, and TA may have also influenced SCL while the students were partaking the quizzes by activating individuals’ BIS. In Study 2, this mediating effect was not significant; however, SA and TA may have affected students’ attention levels when watching micro-lectures and taking quizzes by activating the BIS. The reason for the indirect effect of anxiety on cognitive load not being significant in Study 2 may be due to the differences between the two samples, as excellent students are more adept at handling a given task and reducing their cognitive load [62], whereas students of average ability may complete the task with a potentially greater cognitive load [63]. This suggests an important role for the BIS in the relationship between anxiety and cognitive load and attention levels. Researchers have found that the BIS/BAS can provide an explanatory framework for the relationship between motivation and behaviors [64,65]. According to Gray’s theory, the BIS is closely related to negative emotions [27]. In addition, the BIS/BAS involves motivational tendencies [66], which have been shown by researchers to have a relationship with students’ learning [28,67]. For students who are in a competitive atmosphere every day, the risk of failure in comparison with others is always present, so they feel anxious about their academic performance, and when this emotion reaches a certain level it, activates the BIS, which, in turn, affects the individual’s cognitive load and attention level. This result reminds our educators to pay attention to students’ emotions and motivation in practice and to minimize unnecessary comparisons so that students can focus more on the learning process itself.

## 7. Limitations and Future Research

The several limitations of this study need to be noted. First, in order to strictly control the experimental variables, we used micro-lectures rather than actual online classroom instruction. Future research could explore this in an actual online classroom, but it is important to control for the effects of other extraneous variables on the results. Second, due to some limited experimental conditions, the sample size was not large enough in our studies, especially in Study 2. To improve the accuracy of the conclusions drawn, future studies may consider increasing the sample size or using alternative research methods to validate the findings of the study. Third, some of the results of this study were measured using a self-statement scale, and more objective measurement methods could be used in the future. Fourth, while the focus of this study was on group differences between the competition group and control group, it would be worthwhile to investigate why different students do better or worse in the same competitive environment. As a final point, our studies simply compared data from different groups, and subsequent research could systematically explore the effects of competition on learning in different groups, such as students of different abilities or students from different cultural backgrounds.

## 8. Conclusions

This micro-lecture-based study showed that higher anxiety leads to higher SCL and lower attention through the BIS, suggesting that the BIS’ functioning plays an important role in mediating the effects of anxiety on cognitive load and attention. Moreover, competition seems to benefit the learning performance of lower-ability students but hurt the learning of higher-ability students, which implies that competitive strategies should be used more carefully in educational situations, considering students’ aptitudes.

## Figures and Tables

**Figure 1 behavsci-14-00324-f001:**
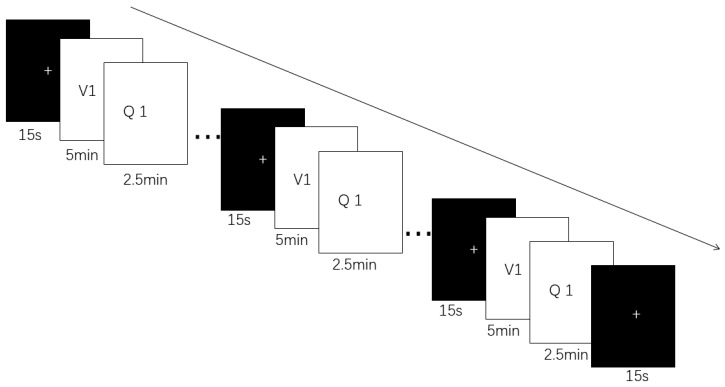
Formal experimental procedure. A 15 s break is shown by the black screen with a white cross on it; Q and V stand for quiz and video.

**Figure 2 behavsci-14-00324-f002:**
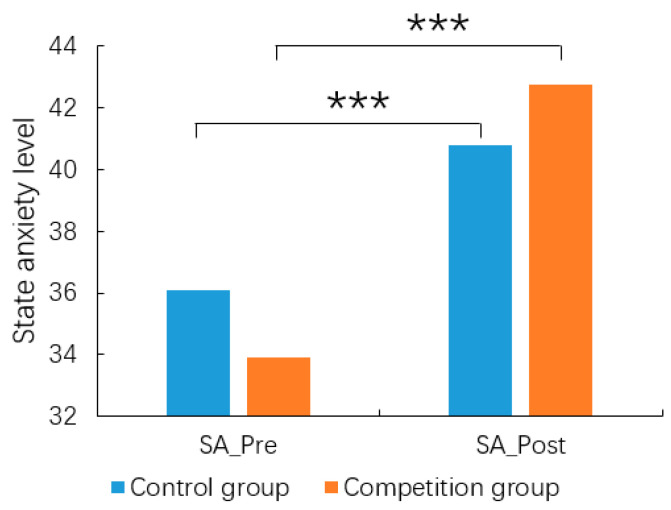
The state anxiety of the competition and control groups before (SA_Pre) and after (SA_Post) the experiment in Study 1. *** *p* < 0.001.

**Figure 3 behavsci-14-00324-f003:**
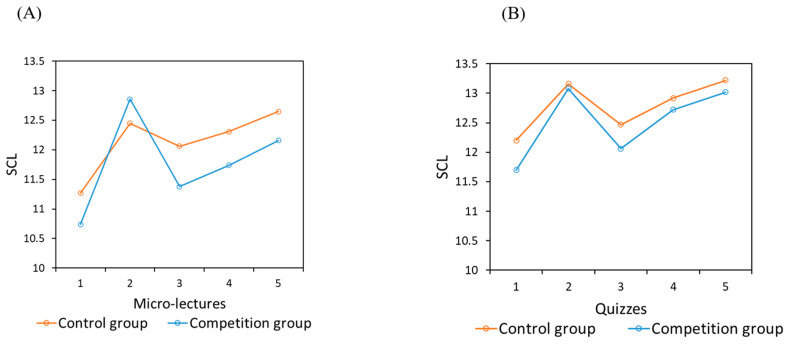
SCL during the five micro-lectures (**A**) and five quizzes (**B**) in Study 1.

**Figure 4 behavsci-14-00324-f004:**
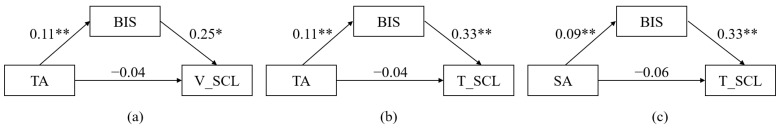
The mediating effect of BIS in Study 1. (**a**): The indirect effect values of the BIS in predicting SCL during watching micro-lectures by TA; (**b**): The indirect effect values of the BIS in predicting SCL during participation in quizzes by TA; (**c**): The indirect effect of values of the BIS in predicting SCL during participation in quizzes by SA. * *p* < 0.05, ** *p* < 0.01.3.3. Discussion.

**Figure 5 behavsci-14-00324-f005:**
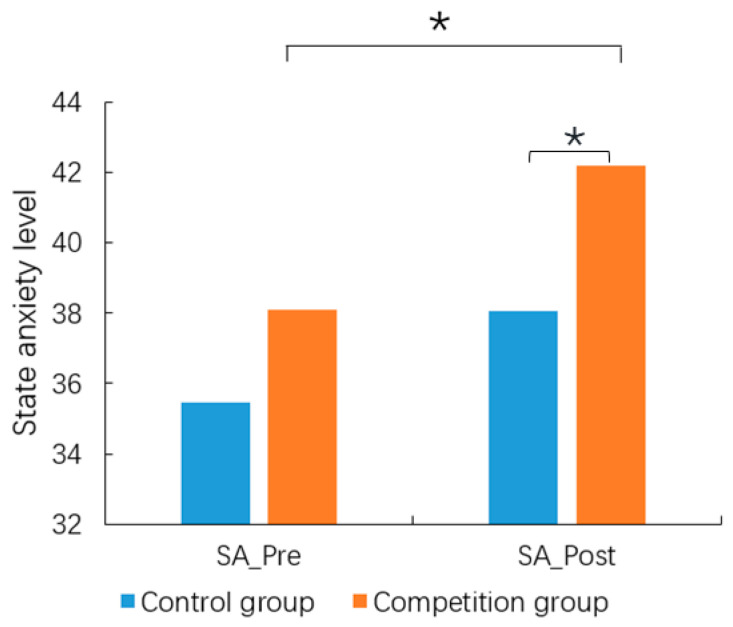
The state anxiety of the competition and control groups before (SA_Pre) and after (SA_Post) the experiment in Study 2. * *p* < 0.05.

**Figure 6 behavsci-14-00324-f006:**
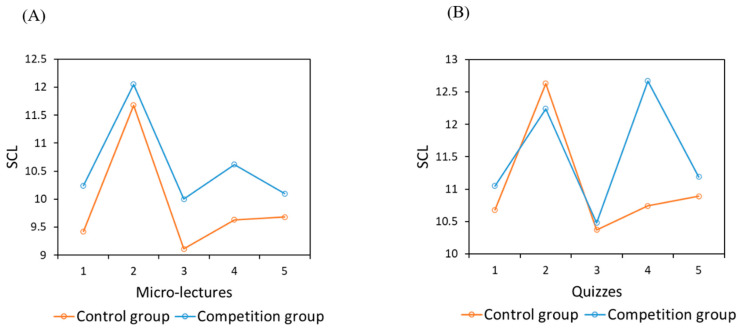
SCL during the five micro-lectures (**A**) and five quizzes (**B**) in Sthdy 2.

**Figure 7 behavsci-14-00324-f007:**
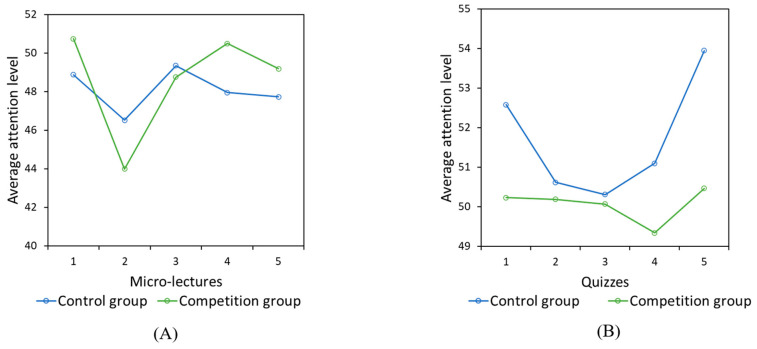
The average attention level during the five micro-lectures (**A**) and five quizzes (**B**).

**Figure 8 behavsci-14-00324-f008:**
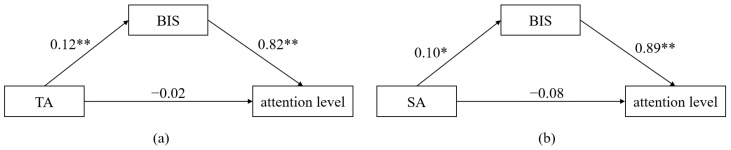
The mediating effect of BIS in Study 2. (**a**): The indirect effect values of the BIS in predicting attention level by TA; (**b**): The indirect effect values of the BIS in predicting attention level by SA. * *p* < 0.05, ** *p* < 0.01.

**Table 1 behavsci-14-00324-t001:** Sample characteristics of competition and control groups in Study 1.

Variables	Competition Group (N = 50)	Control Group (N = 51)	*df*	*χ*^2^ or *t* Value	*p*
Gender (male/female)	6/44	6/45	1	0.001	0.971
Age	20.36 ± 2.14	21.16 ± 2.09	99	1.894	0.061
Years of education	13.82 ± 1.87	14.22 ± 1.89	99	1.057	0.293

**Table 2 behavsci-14-00324-t002:** The relationship between the BIS/BAS and anxiety, SCL, test score.

Variables	BIS		BASR		BASD		BASF	
	*r*	*p*	*r*	*p*	*r*	*p*	*r*	*p*
SA_Pre	0.319 **	0.001	−0.078	0.438	−0.240 *	0.016	−0.106	0.296
SA_Post	0.133	0.183	−0.158	0.114	−0.172	0.086	−0.202 *	0.043
TA	0.365 **	0.000	−0.023	0.817	−0.282 **	0.004	−0.012	0.903
V_SCL	0.188	0.06	0.061	0.544	0.053	0.602	−0.058	0.566
T_SCL	0.260 **	0.009	0.088	0.382	0.056	0.576	0.009	0.925
Test score	0.065	0.517	−0.068	0.499	0.149	0.137	0.006	0.956

Note: SA_Pre = pre-experiment state anxiety, SA_Post = post-experiment state anxiety, TA = trait anxiety, V_SCL = subjective cognitive load in the micro-lectures, T_SCL = subjective cognitive load in the quizzes. * *p* < 0.05, ** *p* < 0.01.

**Table 3 behavsci-14-00324-t003:** Sample characteristics of competition and control groups in Study 2.

Variables	Competition Group (N = 22)	Control Group (N = 20)	*df*	*χ*^2^ or t Value	*p*
Gender (male/female)	3/19	7/13	1	2.636	0.104
Age	21.54 ± 1.53	22.60 ± 2.06	40	1.891	0.066
Years of education	15.50 ± 1.53	16.30 ± 1.98	40	1.472	0.149

**Table 4 behavsci-14-00324-t004:** The relationship between the BIS/BAS and anxiety, attention level, SCL, and test score.

Variables	BIS		BASR		BASD		BASF	
	*r*	*p*	*r*	*p*	*r*	*p*	*r*	*p*
SA_Pre	0.373 *	0.016	−0.04	0.803	−0.242	0.127	−0.121	0.453
SA_Post	−0.077	0.638	0.12	0.461	−0.095	0.561	−0.066	0.066
TA	0.466 **	0.002	−0.249	0.117	−0.331 *	0.035	0.149	0.354
mean_attention	0.464 **	0.004	−0.195	0.248	0.178	0.292	0.262	0.117
V_SCL	0.102	0.535	−0.24	0.141	−0.301	0.063	−0.014	0.935
T_SCL	0.147	0.372	−0.225	0.141	−0.359 *	0.025	−0.001	0.996
Test score	−0.194	0.224	0.053	0.743	0.135	0.398	0.065	0.688

Note: SA_Pre = pre-experiment state anxiety, SA_Post = post-experiment state anxiety, TA = trait anxiety, V_SCL = subjective cognitive load in the micro-lectures, T_SCL = subjective cognitive load in the quizzes. * *p* < 0.05, ** *p* < 0.01.

## Data Availability

We promise that the data will be made available upon receipt of a request and permission to reproduce material from other sources.

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
