# Peer review of "The Effects of Peer Competition-Induced Anxiety on Massive Open Online Course Learning: The Mediating Role of the Behavioral Inhibition System"

_behavsci, 2024, doi:10.3390/bs14040324_

Round 1
Reviewer 1 Report
Comments and Suggestions for Authors
Dear authors
I've carefully read the manuscript, in my opinion, there are several limitations that concerned me:
1. The study rationale in my opinion is measleading. I totally agree with the authors that 'Competition' is somehow part of everyone life, but this is limited to some specific fields (sports, for sure, or work, for example when you are competing for a job or for a promotion). However, in the learning context, I miss the meaning and the place for competition, since there is no a "winner" and a "loser" while learning.
All the students can actually achieve the best performance, without the need for someone else to lose. When learning, the competition is somehow within ourselves (e.g., I wish to achieve - not against someone). Therefore the study rationale should be better justified and theoretically grounded in the context of education and learning.
2. Research justifying the study hypothesis and the methodological approach should be included (mainly mediating hypotheses).
3. Related to the previous points, in my opinion, the use of both state and trait anxiety is a strength of the study but it should be used in a different manner (both theoretically and statistically). Indeed, trait anxiety (as personality trait) could hinder (paralyzing) attention and performance but it can also play a positive role - it depends on the levels. Also, the increase in state anxiety is a normal trend while performing. To justify their conclusions, the authors are suggested to use clinical cut-off points rather than continuous variable. If clinical levels or trait anxiety are present, different conclusions and implications should be commented. If clinical levels of state-anxiety are present (before or after the MOOC), other conclusions and implications should be commented. Also, when a student possess clinical levels of trait anxiety, he/she is more likely to report increase in state anxiety. Therefore, also the use of trait anxiety should be better considered (maybe as control variable?).
3. Minor comments: There are some typos (e.g. Gary/Gray; some authors with lowecases) and there is a need for more clarity throughout the paper.
Comments on the Quality of English LanguageMinor revision suggested.
Reviewer 2 Report
Comments and Suggestions for Authors
General
The study fits within the thematic scope of the journal and is important for scientific development in the field. The considerable effort required to conduct a study of this nature is commendable. Congratulations to the team. However, I would like to offer some suggestions for improvement on certain issues for the research team's consideration:
Introduction There seems to be a confusion between the natural state of competition inherent in humans, which is not universally shared, and the unnatural need to compete imposed by our society (e.g., the example of bonobos) (lines 27-31). Competing is not the only natural and healthy way to interact; it is just one part of it. Information about the risk of suicide due to competitive stress experienced by some students in certain countries should be added, i.e., its impact on mental health, as well as a brief international review on competitiveness. In this regard, the state of the art is limited.
Method The pilot experiment is so briefly summarized that its purpose and process are not clearly understood. It is unclear whether the same students were involved in both the pilot study and the main experiment. Are the students from similar socio-cultural backgrounds? It is not understood why the students are not divided between those who perform well under pressure and those who do not. A prior study in this regard should have been conducted (this point should be addressed in the limitations section). Self-reports are not the most objective way to measure issues like this, and this should be considered to avoid repeating this weakness.
Conclusions The conclusions are very limited and do not capture the essence of what a conclusion should be for a study of this magnitude. Although it is mentioned in the limitations that the sample size is small, the only sentence in the conclusions speaks in general terms. What are the Research Ethics Board protocols of Ningbo University? A paragraph on this would be appropriate.
Also, where are the DOIs?
More up-to-date references should be included.
Round 2
Reviewer 2 Report
Comments and Suggestions for Authors
Dear Editors,
The authors of the current proposal have diligently addressed all the feedback provided. Therefore, I hereby give my approval for its publication.
Kind regards,
Elena
Author Response
Dear reviewer,
Thank you for your letter and approval of the publication of our manuscript, your comments were very helpful in revising and improving the paper, and have been an important guide for our research.
Thank you again for your attention and valuable time.